# The impact of Cochrane Reviews that apply network meta-analysis in clinical guidelines: A systematic review

Sarah Donegan[1]*, James Connor[1], Zarko Alfirevic[2], Catrin Tudur-Smith[1]

**1** Department for Health Data Science, University of Liverpool, Liverpool, United Kingdom, **2** Department of Women and Children's Health, University of Liverpool and Liverpool Women's Hospital, Liverpool, United Kingdom

\* sarah.donegan@liverpool.ac.uk

**Data Availability Statement:** All data and R code are available in the supplementary material. The original protocol, which was unregistered, can be obtained from the corresponding author.

## Abstract

### Background

Systematic reviews, such as those prepared by Cochrane, are the strongest evidence on which to base recommendations in clinical guidelines. Network meta-analysis (NMA) can be used to combine the results of studies to compare multiple treatments, which is advantageous over pair-wise meta-analysis (PW-MA) that compares two treatments. We aimed to summarise which, when, where, who, and why Cochrane Reviews that applied NMA were cited in guidelines; and to compare the citation of NMA reviews in guidelines with PW-MA reviews.

### Methods and findings

We carried out a systematic review of Cochrane reviews that applied NMA and we summarised their citation in guidelines. The Cochrane Database of Systematic Reviews was searched (15th January 2024). Additionally, a cohort of Cochrane reviews that applied PW-MA was matched to the NMA reviews. Two authors assessed eligibility and extracted data. We summarised review and guideline characteristics, and the use of the review in guidelines.

### Results

Of the 60 included NMA reviews, 26 reviews (43%) were cited in 89 guidelines (1–13 per review). 15 NMA reviews (58%) were first cited within two years of publication, with the remaining 11 reviews (42%) cited 2–6 years later. 52 guideline developers authored the guidelines. The number of citations was higher for NMA than PW-MA reviews (rate ratio 1.53 (1.08 to 2.19), p = 0.02). The number of times reviews were commissioned or cited alongside a recommendation was also higher for NMA than PW-MA reviews (rate ratio 4.40 (1.80 to 13.14), p = 0.003). NMA reviews were more likely to be cited in the text surrounding a recommendation or used for NICE guideline development (1.94 (1.08 to 3.63), p = 0.03).

**Funding:** The author(s) received no specific funding for this work.

**Competing interests:** I have read the journal's policy and the authors of this manuscript have the following competing interests: CTS and SD were statistical editors for Cochrane Review Groups, ZA was co-ordinating editor for the Cochrane Pregnancy and Childbirth Group, and CTS is a co-convenor for the Cochrane Statistical Methods Group.

## Conclusions

Cochrane NMA reviews appear to have more impact than PW-MA reviews, but many are not cited in guidelines. Further work is needed to explore the barriers to use of NMAs and promote their use.

## Introduction

Systematic reviews of randomised controlled trials provide the strongest evidence to underpin healthcare decisions and can be used by guideline developers, such as the National Institute for Health and Care Excellence (NICE), to produce clinical guidelines that state treatment recommendations that health care providers follow to deliver patient care [1–5]. Cochrane reviews published in the Cochrane Database of Systematic Reviews (CDSR), are high quality systematic reviews that apply the latest, rigorous methods set out by Cochrane [3, 6]. International standards and guidance for guideline developers require that a systematic review is carried out as part of the guideline development process, in fact, the World Health Organisation guidance specifies that Cochrane methods should be followed [7–11]. Nevertheless, Cochrane Reviews have been greatly under-utilised in guidelines and recommendations are often based on lower-level evidence or expert consensus because of limitations regarding familiarity, awareness, usefulness, access, and time [12–21]. Review authors strive to maximise the impact of their review in terms of improvements to patient heath [22]. Yet, if a review does not provide evidence to strengthen or change recommendations or is not used by guideline developers, the review could potentially have very limited impact on healthcare and may be considered a poor use of time, funds, and resources.

When reviews aim to compare multiple treatments (e.g. *A*, *B*, *C*) for a particular disease and specified population, review authors can use network meta-analysis (NMA) that combines the trials' results to rank the treatments in order of effectiveness or safety, and to estimate the relative treatment effects for all treatment comparisons (e.g. odds ratio for *B* vs. *A*, odds ratio for *C* vs. *A*, odds ratio for *C* vs. *B*) using direct and indirect evidence [3, 23–28]. Direct evidence for comparison *C* vs. *B* is from head-to-head trials of *C* vs. *B*; whereas indirect evidence would come from head-to-head trials of *B* vs. *A* and head-to-head trials of *C* vs. *A* [23]. NMA is an extension of pairwise meta-analysis (PW-MA) that can compare only two treatments [3]. There for two underlying assumptions of NMA: homogeneity, that also underlies PW-MA, and consistency [3, 29–32]. Unreliable results may be produced when either assumption is unreasonable [3, 24]. NMAs have been described as the highest level of evidence, above PW-MAs, and have huge potential to provide important evidence to inform guidelines [31, 33–37]. When no direct evidence is available for a particular treatment comparison, the relative treatment effect will still be estimated by NMA using indirect evidence enabling recommendations to be made, whereas no result would be produced from PW-MA. Also, as NMA is generally based on more trials and patients than PW-MA, the results produced by NMA are often more precise than those from PW-MA enabling more reliable recommendations to be made. Nonetheless, the National Institute for Health and Care Research recommend their use only when direct evidence is lacking [10]. However, the validity of direct and indirect evidence will depend on the risk of bias of studies and feasibility of the assumptions, such that in some cases, the direct evidence is preferred, and in other cases, indirect evidence is favourable [28].

Previous work has reviewed guidelines and summarised the type of evidence that underpinned recommendations [12, 13, 15]. Limited research has explored the use of NMAs to

inform guidelines [38]. Lunny *et al* summarised how often NMAs and PW-MAs support recommendations in a sample of 50 guidelines [15]. Our article is the first to systematically search for reviews and summarise their citation in guidelines, and to explore the citation of reviews in depth. In this article, we build on prior research by conducting a systematic review of Cochrane Reviews that applied NMA to summarise which reviews were cited in guidelines in terms of review characteristics, when the reviews were cited, where the reviews were cited regarding location, who cited the reviews, and why the reviews were cited in terms of how the review contributed to the guideline.

Hypothetically, NMA reviews may be less likely to be cited in guidelines than PW-MA reviews because of lack of understanding or cautiousness regarding NMA methods. Previously, in a review of guidelines, Lunny *et al* found that NMA reviews are less cited than PW-MA reviews [15]. However, to our knowledge, no research has reviewed PW-MA reviews and summarised their citation in guidelines or compared samples of NMA reviews and PW-MA reviews in terms of citation. As a post hoc sub-study, a cohort of Cochrane reviews that applied PW-MA is matched to the NMA reviews to compare their citation in guidelines.

We anticipate that our research will highlight the extent of potential research waste, help review authors to maximise the impact of their reviews, aid guidelines developers to improve and increase the use of NMA and PW-MA reviews, and lead to improved collaboration to increase research efficiency.

## Methods

Ethical approval was not sought or required because this research involved reviewing only published research.

### Eligibility criteria for NMA reviews

Cochrane Reviews of interventions that applied NMA were included. We excluded protocols, withdrawn reviews, overview of reviews, and diagnostic test accuracy reviews. We excluded reviews that used indirect comparison methods (e.g. calculating the difference between results obtained from pairwise meta-analyses) rather than NMA models because indirect comparisons are made in reviews without using searchable terminology. We excluded reviews that applied NMA but for which the main summary of findings table was based on PW-MA or single trials because we primarily aimed to explore the uptake of NMA results in clinical guidelines.

### Search for NMA reviews and assessment of their eligibility

The CDSR was searched for NMAs to 15th January 2024 (see S1 File for search strategy). Reviews were filtered to include only intervention reviews.

The publication of each review was obtained. SD assessed the eligibility of reviews using a form. JC independently assessed a subset of 70 reviews (55%) and differences were resolved.

### Search for guidelines cited by NMA reviews

Cochrane UK continually search the national guideline portals and websites of the UK, Australia, USA, Canada and Europe; the Guideline International Network portal; World Health Organization guidelines; and PubMed for guidelines. They search each guideline for the word "Cochrane" and when a Cochrane Review is cited, they enter bibliographic details for the guideline into their database in the Cochrane Register of Studies. Wiley publishes the references to guidelines that cite each review in CDSR. We sought the publication of each guideline associated with the included NMA reviews.

## Data extraction of NMA reviews

SD extracted data from the included reviews, their associated guidelines, and summary information in CDSR regarding guidelines using a form. JC independently extracted data for 58 reviews (97%) and differences were discussed.

We extracted the review citation, Cochrane Review Group, publication date, sources of support, types of studies and participants, the interventions in the network for the first outcome (i.e. the first outcome in the methods for which data were collected and NMA results were reported). For the first outcome, we extracted whether it was a primary or secondary outcome, the type of data, measure of effect used, the number of trials and patients analysed, whether heterogeneity or inconsistency were found and assessment methods, whether a frequentist or Bayesian approach was followed, grade ratings, and the risk of bias (for sequence generation and allocation concealment) of the studies included in the analysis. When multiple analyses were carried out (e.g. for drug classes and individual drugs) we extracted data for the first reported analysis.

For each guideline, we recorded the publication year, location, authors, title, citation, and how the review was used in the guideline. We defined four levels of impact of a review on the guideline:

- Level one: the guideline stated that the review was commissioned to provide evidence, or the review was cited alongside a recommendation in a guideline (e.g. a review was cited in the same sentence as a clear recommendation).

- Level two: the review was used as part of NICE guidance development (e.g. contributing supporting evidence), or the review was cited in the text surrounding a recommendation in a guideline (e.g. alongside other evidence, an NMA was described and cited in the paragraph preceding or following a clear recommendation).

- Level three: a review was cited without making recommendations (e.g. an NMA was described and cited, but no recommendation surrounded the description).

- Level four: the review was listed as a reference in the bibliography but not cited in the guideline.

## Data analysis of NMA reviews

The characteristics of reviews were tabulated. Continuous data were presented as medians and ranges, and numbers and percentages were displayed for categorical data. Histograms were used for the publication year, time to first citation, the number of times reviews were cited, and why reviews were cited. A map was used to present the location of guideline groups. A Kaplan Meier plot was presented for time to citation, which was calculated by subtracting the publication year of the review from the year that the review was first cited in a guideline. Reviews that were not cited were censored and their time to citation was calculated by subtracting the publication year of the review from the current year (2024). All figures were produced in Excel except Kaplan Meier plots which were constructed in R 4.4.0. Non-reported missing data were excluded from analyses.

## Sub-study comparing the citation of NMAs and PW-MA reviews in guidelines

**Eligibility of PW-MA reviews.** We matched each included NMA to a PW-MA published by the same Cochrane Review Group, and with the closest publication date to the NMA (but

with no more than a one-month difference between the publication dates of the NMA and PW-MA). Eligible PW-MA reviews compared interventions, applied PW-MA, were not withdrawn, and did not describe indirect comparison or NMA methods.

**Search for PW-MA reviews and assessment of their eligibility.**   For each included NMA, SD searched the CDSR, filtering by Review Group, to locate reviews published the same month as the NMA and in the preceding and following month. Located reviews were assessed for eligibility starting with the review published closest to the publication date of the NMA, and then to those increasingly further in time to a maximum of one month before or after the NMA publication date. In cases were the nearest eligible PW-MA was already matched to another NMA, we matched the next nearest eligible PW-MA. Reasons for exclusion of reviews were documented.

**Data extraction and analysis of PW-MA reviews.**   The search for guidelines and data extraction and analysis methods were analogous to those described for NMA reviews, except only SD carried out assessments and extraction. We also compared the impact of NMAs and PW-MA reviews, by fitting regression models with review type (NMA or PW-MA) as an independent variable using the glymer, glm and coxph functions in R.4.4. A logistic regression model was fitted for the binary variable regarding whether a review was cited in guidelines. Poisson regression models were used for count variables regarding the number of times reviews were cited in guidelines and the number of level one, level two, and level three impacts in guidelines. A cox proportional hazards regression model was applied for time to first citation in guidelines. PW-MA reviews were the baseline in models such that effect estimates (i.e. odds ratio, rate ratio, and hazard ratio) compare NMA versus PW-MA reviews. Models that accounted for matching by including a random effect for each matched pair were fitted as secondary analyses because there is debate regarding whether adjustment for matching is appropriate for cohort studies [39]. R code is provided in S4 File.

## Results

### NMA review profile

Fig 1 displays the eligibility assessment process. S2 File provides the references to the excluded reviews and reasons for exclusion. Sixty NMA reviews were included. S1–S3 Tables display the data extracted and provides references.

### Which NMA reviews are cited?

Twenty-six reviews (43%) were cited in guidelines.

Older reviews were more frequently cited in guidelines than newer reviews (Fig 2). For some Cochrane Review Groups, all reviews were cited in guidelines, whereas no review was cited for other groups (Table 1). The average number of patients was higher for reviews that were not cited (median 6,270 (range 458–145,460)) than those that were cited (median 3,851 (range 737–50,812)) (Table 2). Reviews with lower grade ratings may be less likely to be cited: the lowest grading of the results in reviews that were cited was very low for 12 reviews (40%), low for seven reviews (35%), moderate for two reviews (67%) and high for one review (100%) (Table 2).

There were no obvious differences between cited and no-cited reviews regarding sources of support, types of studies, the number of trials and interventions, type of analysis, presence of heterogeneity or inconsistency, and risk of bias of the included studies (Tables 1 and 2).

Reviews were cited in guidelines even when heterogeneity (13 reviews) or inconsistency (five reviews) was present, grade ratings were very low (12 reviews), or risk of bias was high (four reviews).

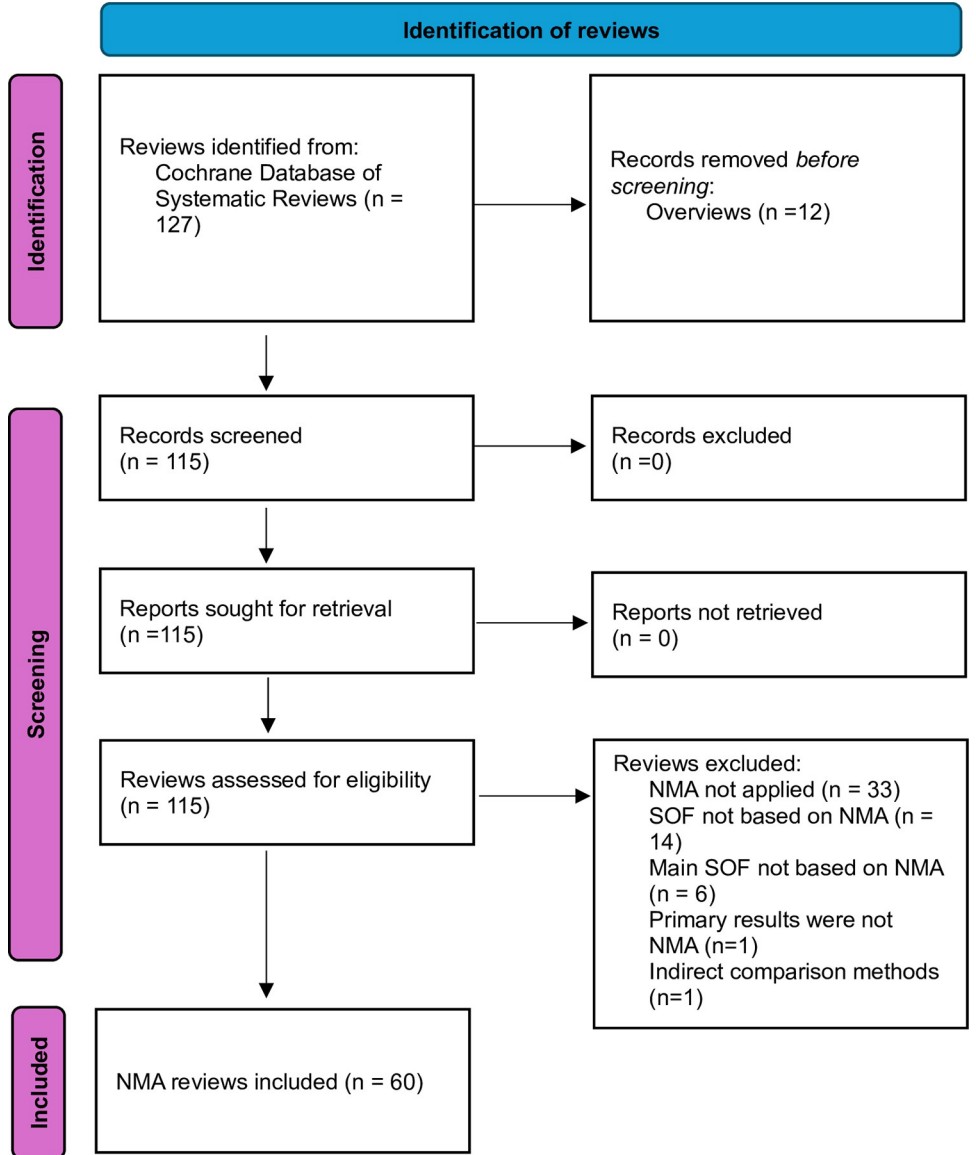

**Fig 1. Flow diagram of NMA reviews.** NMA: network meta-analysis; SOF: summary of findings.

## When are NMA reviews cited?

Of the 26 reviews that were cited, 15 reviews (58%) were first cited within two years post-publication, with the remaining 11 reviews (42%) first cited up to six years later (Fig 3). Thirty-four reviews (57%), published between 2017 and 2024, have not yet been cited in guidelines. Fig 4 shows that 75% of reviews were not cited after two years, and 65% of reviews were not cited even six years later.

## Where are NMA reviews cited?

The 26 reviews were cited in 120 guidelines. Some reviews were cited by multiple versions of the same guideline and 89 guidelines remained when only the most recent version of the guideline was retained. The 26 reviews were each cited by between 1 and 13 guidelines, with

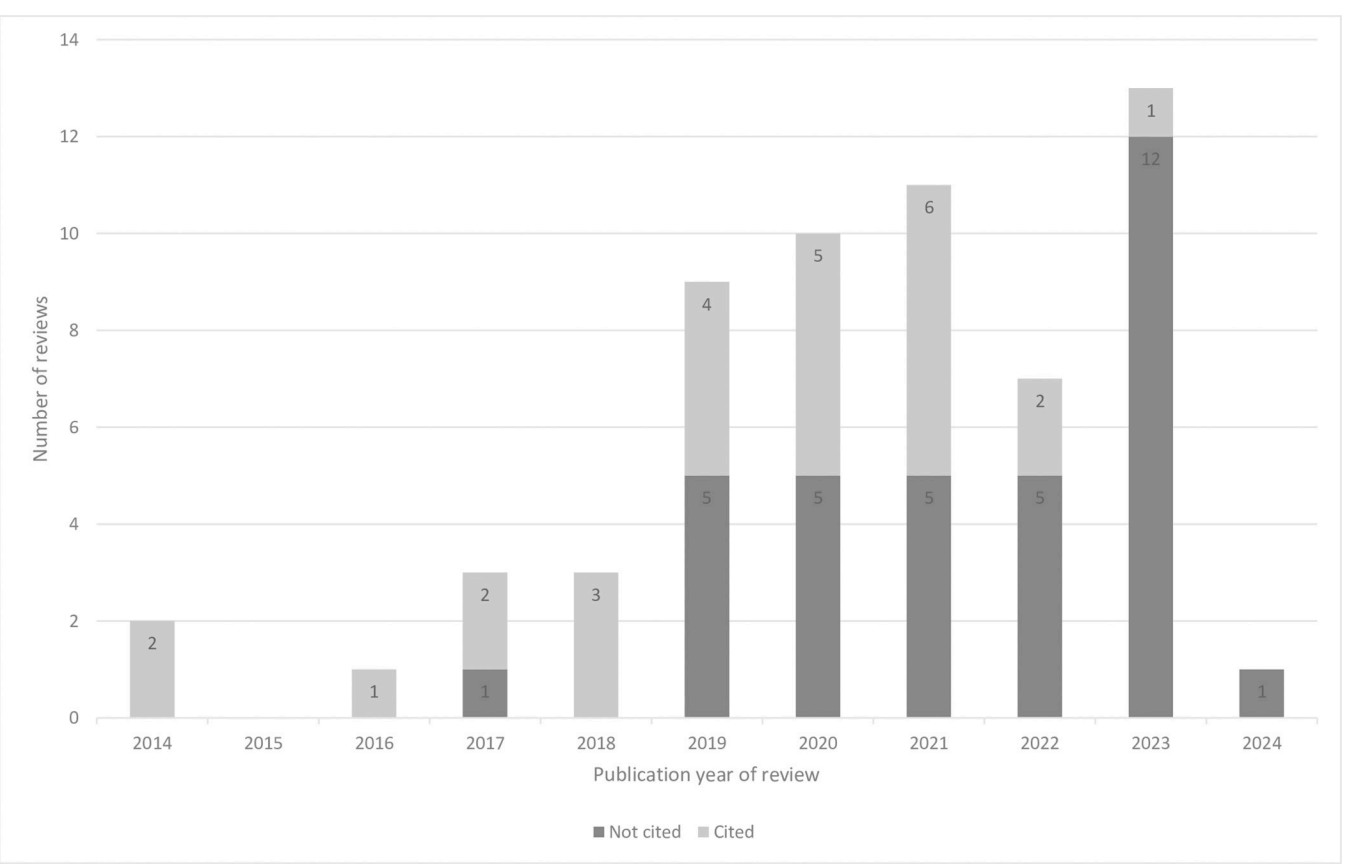

**Fig 2. Number of NMA reviews cited in guidelines and those not cited, for each publication year.**

an average of two guidelines (Fig 5). The 89 guidelines were developed by groups located globally (Fig 6).

## Who cites NMA reviews?

Table 3 shows the 52 guideline development groups of the 89 guidelines. Fifteen guidelines (17%) were developed by National Institute for Health and Care Excellence and seven guidelines (8%) by the World Health Organization. The remaining groups developed three guidelines (four groups), two guidelines (nine groups), or one guideline (37 groups) (Table 3).

## Why are NMA reviews cited?

Reviews contributed to guidelines to various extents (Fig 7). Of the 26 cited reviews, the highest impact in any guideline was level one for 16 reviews (62%), level two for seven reviews (27%), and level three for three reviews (12%) (Fig 7).

Reviews had a level one impact in 27 of the 89 guidelines (30%): four guidelines (4%) describe commissioning reviews to provide evidence, and 23 guidelines (26%) used a review as the evidence source for a recommendation. There was a level two impact in 34 of the 89 guidelines (38%): 10 guidelines (11%) used reviews for NICE guideline development, and 24 guidelines (27%) referenced a review in the text surrounding a recommendation. Twenty-three guidelines (26%) referenced a review but without making recommendations, that is a level

**Table 1. NMA review characteristics.**

| Characteristic | | Number of reviews not cited in guidelines (N = 34) | Number of reviews cited in guidelines (N = 26) | Total |
|---|---|---|---|---|
| Review Group (n (%)) | Airways | 2 (40) | 3 (60) | 5 |
| | Anaesthesia | 0 (0) | 1 (100) | 1 |
| | Bone, Joint & Muscle Trauma | 2 (100) | 0 (0) | 2 |
| | Breast Cancer | 1 (100) | 0 (0) | 1 |
| | Common Mental Disorders | 1 (33) | 2 (67) | 3 |
| | Developmental, Psychosocial & Learning Problems | 1 (100) | 0 (0) | 1 |
| | Emergency & Critical Care | 0 (0) | 1 (100) | 1 |
| | Epilepsy | 0 (0) | 1 (100) | 1 |
| | Eyes & Vision | 2 (67) | 1 (33) | 3 |
| | Fertility Regulation | 0 (0) | 1 (100) | 1 |
| | Gut | 1 (100) | 0 (0) | 1 |
| | Gynaecological, Neuro-oncology & Orphan Cancer | 3 (100) | 0 (0) | 3 |
| | Gynaecology & Fertility | 1 (100) | 0 (0) | 1 |
| | Haematology | 1 (50) | 1 (50) | 2 |
| | Heart | 1 (100) | 0 (0) | 1 |
| | Hepato-Biliary | 3 (33) | 6 (67) | 9 |
| | Incontinence | 1 (100) | 0 (0) | 1 |
| | Kidney & Transplant | 1 (100) | 0 (0) | 1 |
| | Movement Disorders | 0 (0) | 1 (100) | 1 |
| | Multiple Sclerosis and Rare Diseases of the Central Nervous System | 2 (100) | 0 (0) | 2 |
| | Neonatal | 2 (100) | 0 (0) | 2 |
| | Oral Health | 0 (0) | 1 (100) | 1 |
| | Pain, Palliative & Supportive Care | 3 (100) | 0 (0) | 3 |
| | Pregnancy & Childbirth | 0 (0) | 4 (100) | 4 |
| | Skin | 1 (50) | 1 (50) | 2 |
| | Tobacco Addiction | 1 (100) | 0 (0) | 1 |
| | Urology | 3 (100) | 0 (0) | 3 |
| | Work | 0 (0) | 1 (100) | 1 |
| | Wounds | 1 (50) | 1 (50) | 2 |
| Sources of support (n (%)) | Funding body | 15 (58) | 11 (42) | 26 |
| | Funding body and government | 3 (38) | 5 (63) | 8 |
| | Funding body and hospital/network; | 1 (50) | 1 (50) | 2 |
| | Funding body and charity | 1 (33) | 2 (67) | 3 |
| | Funding body, hospitals/network, and government. | 1 (50) | 1 (50) | 2 |
| | Funding body, hospitals/network, and health agency | 0 (0) | 1 (100) | 1 |
| | Funding body, health agency, hospitals/networks, and charity | 0 (0) | 1 (100) | 1 |
| | Funding body, health agency, and government | 1 (50) | 1 (50) | 2 |
| | Health agency | 1 (100) | 0 (0) | 1 |
| | Hospitals/network | 2 (100) | 0 (0) | 2 |
| | Government | 5 (83) | 1 (17) | 6 |
| | Hospital/networks and government | 0 (0) | 1 (100) | 1 |
| | Hospitals/network and charity | 2 (100) | 0 (0) | 2 |
| | Not reported | 2(67) | 1(33) | 3 |
| Studies (n (%)) | RCTs | 30 (57) | 23 (43) | 53 |
| | RCTs (including quasi-RCTs). | 3 (60) | 2 (40) | 5 |
| | RCTs and non-RCTs. | 1 (50) | 1 (50) | 2 |

**Table 2. Characteristics of the NMA for the first outcome.**

| Characteristic | | Number of reviews not cited in guidelines (N = 34) | Number of reviews cited in guidelines (N = 26) | Total |
|---|---|---|---|---|
| **Number of interventions (median, range)** | | 9 (3–24) | 7 (3–65) | - |
| **Number of trials (median, range)** | | 24.5 (4–299) | 22 (3–282) | - |
| **Number of patients (median, range)** | | 6270 (458–145460) | 3851(737–50812) | - |
| **Type of analysis (n (%))** | Frequentist | 22 (56) | 17 (44) | 39 |
| | Bayesian | 8 (67) | 4 (33) | 12 |
| | Frequentist and Bayesian. | 4 (44) | 5 (56) | 9 |
| **Heterogeneity present (n (%))** | Yes | 15 (54) | 13 (46) | 28 |
| | no | 12 (63) | 7 (37) | 19 |
| | Not reported/applied | 7 (54) | 6 (46) | 13 |
| **Inconsistency present (n (%))** | Yes | 7 (58) | 5 (42) | 12 |
| | No | 20 (59) | 14 (41) | 34 |
| | Not reported/applied | 7 (50) | 7 (50) | 14 |
| **Grade classification (n (%))** | Very low for at least one comparison (with other comparisons classed as low, moderate or high) | 18 (60) | 12 (40) | 30 |
| | Low for at least one comparison (with other comparisons classed as moderate or high) | 13 (65) | 7 (35) | 20 |
| | Moderate for at least one comparison (with other comparisons classed as high) | 1 (33) | 2 (67) | 3 |
| | High for all comparisons | 0 (0) | 1 (100) | 1 |
| | Threshold analysis/ not reported | 2 (33) | 4 (67) | 6 |
| **Risk of bias (sequence generation and allocation concealment) (n (%))** | Low | 4 (80) | 1 (20) | 5 |
| | Low and unclear or not reported but all trials in review are unclear or low | 19 (50) | 19 (50) | 38 |
| | One or more high risk trial | 9 (69) | 4 (31) | 13 |
| | Not reported | 2 (50) | 2 (50) | 4 |

three impact. There was a level four impact in two guidelines (2%) which included a review in the bibliography only (Fig 8).

## Whether NMA reviews have a greater impact in guidelines than PW-MA reviews?

**Matching NMA and PW-MA reviews.**    Thirty-eight of the 60 NMA reviews were matched to PW-MA reviews. NMA reviews were not matched because no other reviews were published within one month (11 reviews) or reviews published within one month were ineligible (11 reviews) (S3 File). S4–S6 Tables show the data extracted and provides references.

**Comparison of characteristics of the PW-MA and NMA reviews.**    The NMA and PW-MA reviews differed regarding number of interventions (NMA: average 7 (range 3–25); PW-MA: 2.5 (2–12)), trials (NMA: 22 (3–124); PW-MA: 11 (2–142)) and patients (NMA: 4,167 (737–27,024); PW-MA: 1,615 (159–1,101,795)). The reviews also differed regarding the use of Bayesian methods (NMA: 21% vs PW-MA 0%) and reporting of heterogeneity (NMA: 65% vs PW-MA 32%). The NMA and PW-MA reviews were reasonably similar regarding the included study design, risk of bias; and grade ratings (S7 and S8 Tables).

**Comparison of citation of the PW-MA and NMA reviews.**    Eighteen NMA reviews (47%) were cited in guidelines, whereas 24 PW-MA reviews (63%) were cited. No difference in the likelihood of citation was detected between NMA and PW-MA reviews (odds ratio 0.53,

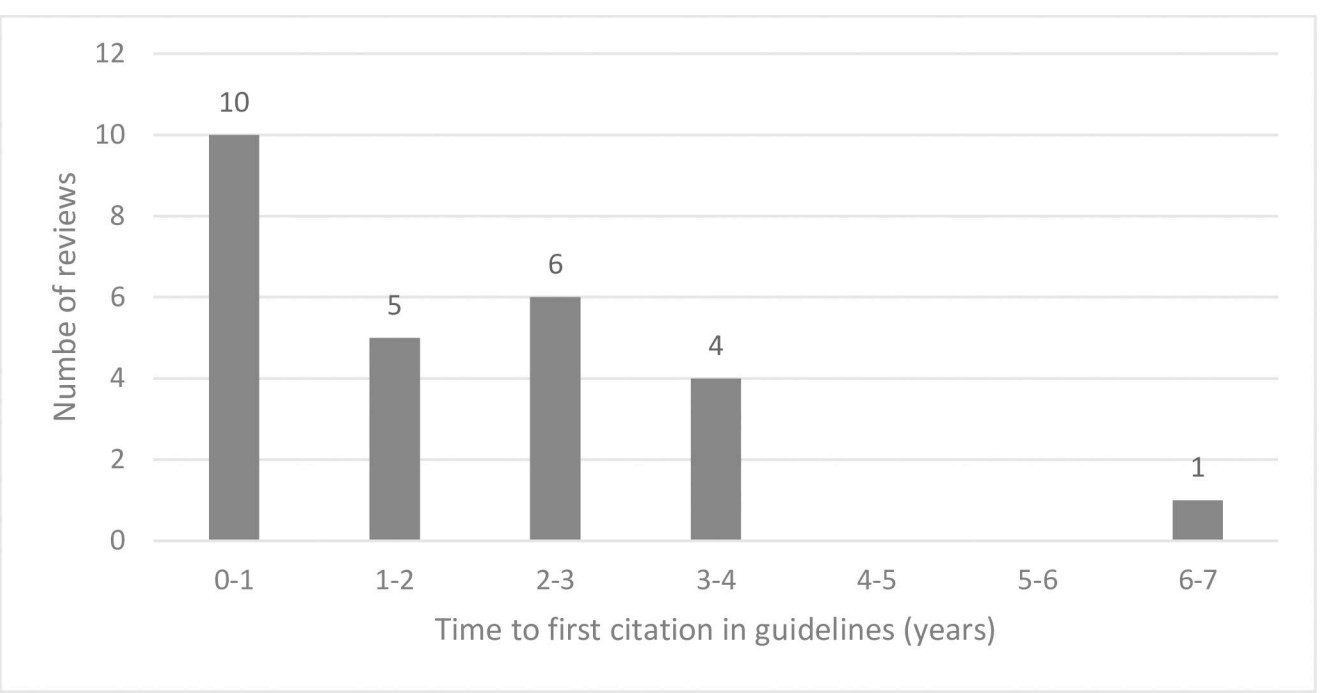

**Fig 3. Time to first citation in guidelines for NMA reviews that were cited.**

CI (0.21 to 1.30), p = 0.17). Results did not change substantially after adjusting for matching (0.37, CI (0.09 to 1.16), p = 0.11). An equal number of PW-MA and NMA reviews were cited each year between 2014–2019 but more PW-MA reviews than NMA reviews were cited between 2020–2023 (S1 Fig).

A similar number of NMA and PW-MA reviews were cited within two years of publication (NMA: 10, 26% vs PW-MA: 11, 29%) and between 2–6 years later (NMA: 8, 21% vs PW-MA: 13, 34%) (S2 Fig). The time to first citation did not differ between PW-MA and NMA reviews (hazard ratio 0.77, CI (0.42 to 1.42), p = 0.40) (S3 Fig).

The number of times NMA reviews were cited was higher than for PW-MA reviews (rate ratio 1.53, CI (1.08 to 2.19), p = 0.02). Adjusting for matching did not change the conclusions (1.53, CI (0.46 to 0.93), p = 0.02). The NMA and PW-MA reviews were cited by 109 guidelines (78 after versions removed) and 69 guidelines (51 after versions removed) respectively. The 24 cited PW-MA reviews were cited by 1–10 guidelines each (average two guidelines); whereas the 18 cited NMA reviews were cited by 1–13 guidelines (average four guidelines) (S4 Fig).

The locations of the guideline developement groups were comparable for PW-MA and NMA reviews (S5 Fig). There was no obvious differences between the groups that cited NMA and PW-MA reviews (S9 Table).

The number of times that NMA reviews had a level one was higher than for PW-MA reviews (rate ratio 4.40 (1.80 to 13.14), p = 0.003); more guidelines associated with NMA reviews than PW-MA reviews were commissioned specifically to provide evidence (NMA: 4, 5% vs PW-MA: 0, 0%) and referred to a review alongside a recommendation (19, 24% vs: 5, 10%) (S6 Fig). The number of times that NMA reviews had a level two impact was higher than for PW-MA reviews (1.94 (1.08 to 3.63), p = 0.03); more guidelines associated with NMA reviews than PW-MA reviews cited a review in the text surrounding recommendations (22, 28% vs 8, 16%) but more guidelines associated with PW-MA reviews than NMA reviews used reviews as part of NICE guideline development (NMA: 8, 10% vs PW-MA: 8, 16%). The

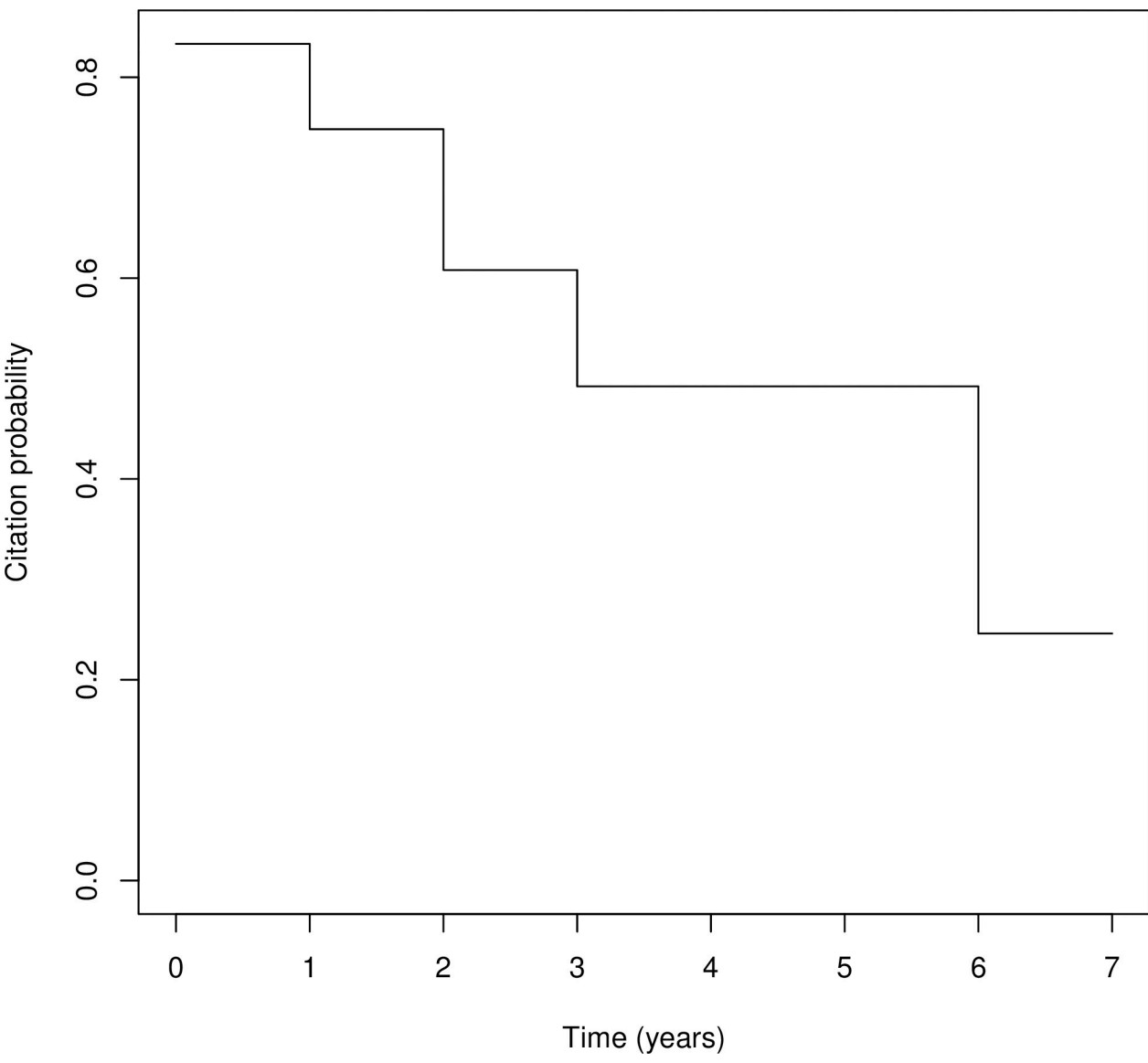

**Fig 4. Kaplan Meier curve of time to first citation in guidelines for NMA reviews.**

number of times that NMA reviews had a level three impact did not differ from that of PW-MA reviews (0.88, CI (0.49 to 1.56), p = 0.66); however more guidelines associated with PW-MA reviews than NMA reviews referenced a review but without making recommendations (20, 26% vs 25, 49%). Accounting for matching did not change the findings (level one: 4.40 CI (1.67 to 11.62), p = 0.003; level two: 1.94, CI (1.06 to 3.54), p = 0.03; level three: 0.88 CI (0.50 to 1.56), p = 0.66).

## Discussion

### Principal findings

We found that reviews appear to be under-utilised in guidelines with only 43% of Cochrane NMA reviews cited in guidelines and 27% clearly underpinning recommendations. Therefore, 57% of NMA reviews potentially have minimal impact on patient health; and addition reviews

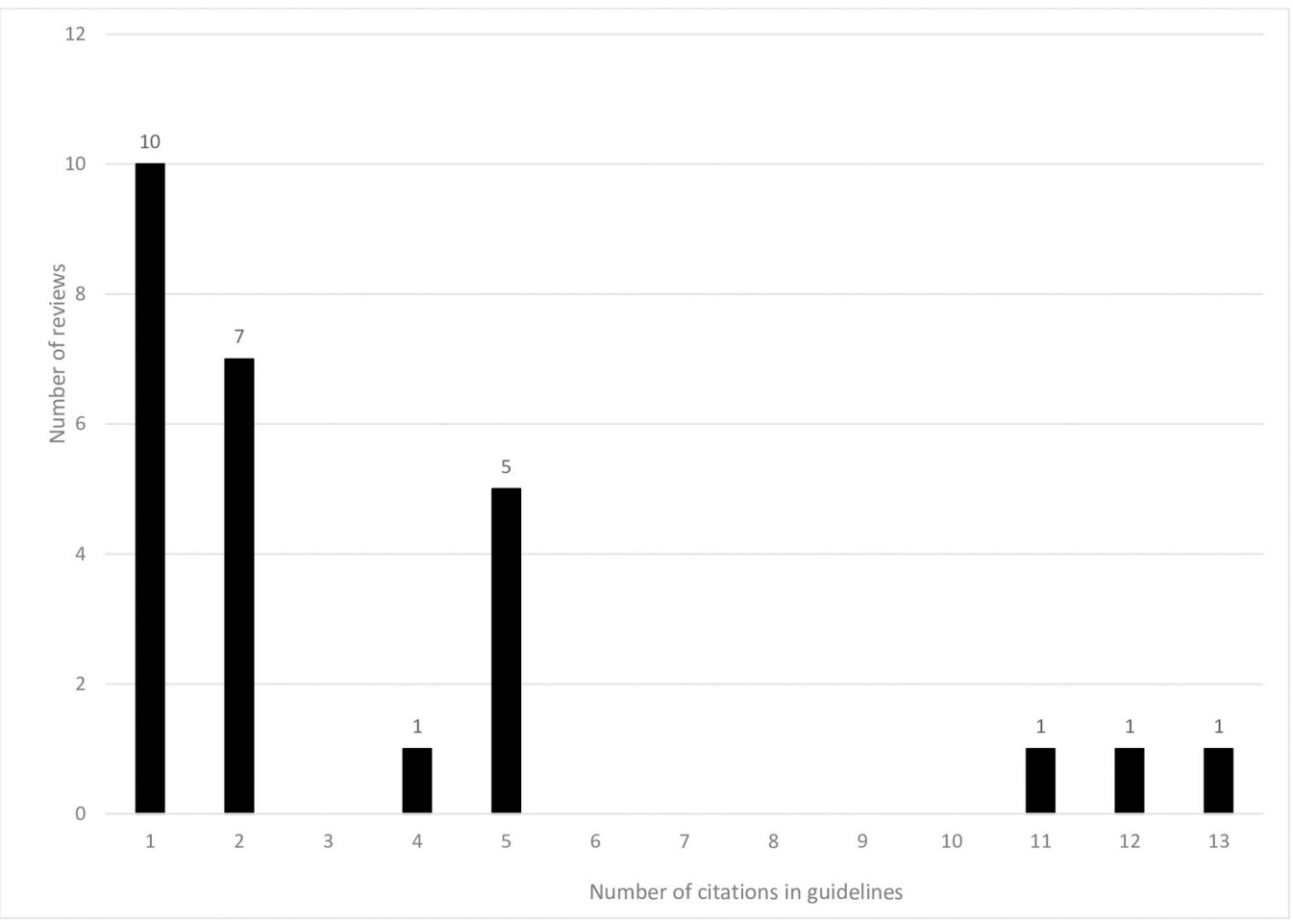

**Fig 5. Number of times NMA reviews were cited in guidelines (excluding NMA reviews that were not cited).**

have been cited in guidelines but not actually changed or strengthened recommendations. Our findings are consistent with previous work. Korfitsen *et al* found that only 18% of 585 recommendations made in 56 guidelines published by the Danish Health Authority between 2014–2021, included evidence from a Cochrane Review [13]. Lunny *et al* found that 249 reviews (of which 64% were systematic reviews using PW-MAs and 3% using NMAs) were cited in 50 guidelines (dated 2017–2018) retrieved from the TRIP and Epistemonikos databases [15]. Reasons for not citing reviews could include guideline developers being unaware of the review; differing research questions; the review not changing or strengthening recommendations; or low credibility of the review's results. Moreover, an NMA review may not be cited because of lack of understanding or cautiousness regarding NMA.

We found that some guidelines cited low quality evidence and out-dated reviews. Of the 26 NMA reviews that were cited in guidelines, 50% reported heterogeneous results, 19% reported inconsistent results, 73% reported low and very low credibility results, and 15% included studies with a high risk of bias. Previously, Vigna-Taglianti *et al* assessed the quality of 80 reviews cited in 128 guidelines regarding breast and colon cancer and found 70% of reviews were low quality reviews [17]. If the findings were not suitably interpreted by the review authors or guideline developers, use of such reviews may lead to inappropriate recommendations. Furthermore, reviews require regular updating to include newly published studies or apply new

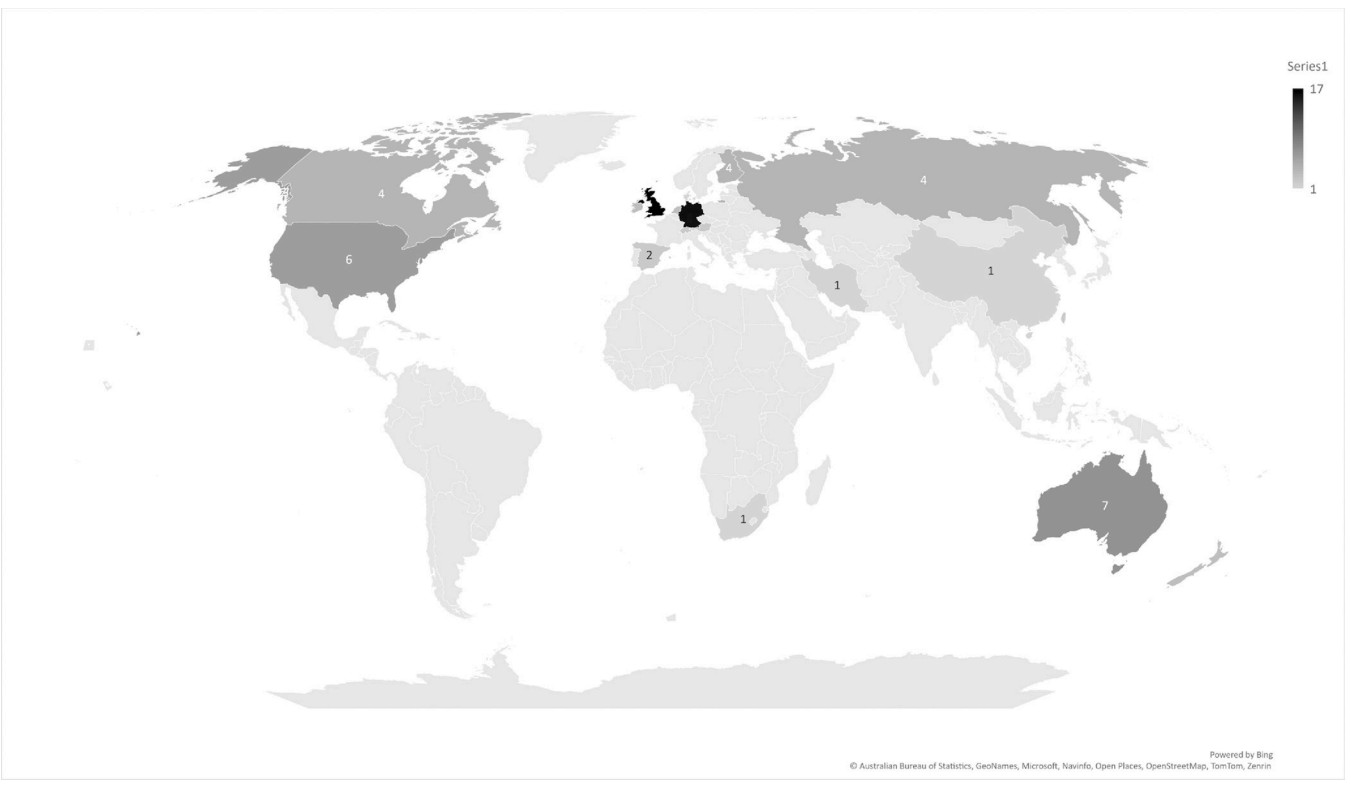

**Fig 6. Location of guidelines that cited NMA reviews.** UK (17 guidelines); Germany (14); Global (10); Europe (7); USA (6); Canada (4); Finland (4); Russia (4); Australia (4); Netherlands (3); Australia and New Zealand (3); Ireland (3); Spain (2); Germany, Austria, Switzerland (2) Taiwan (2); Iran (1); South Africa (1); Denmark (1); China (1).

methods that would alter findings [3]. As general guidance, the Cochrane recommends updating reviews at least every two years [3]. Yet, we found that 42% of cited NMA reviews were first cited between 2–6 years after publication, therefore guidelines may be citing out-dated reviews.

This research showed that NMA reviews have more impact in guidelines than PW-MA reviews. We found that PW-MA and NMA reviews did not differ regarding the likelihood of citation, time to first citation, and location. The number of times NMA reviews were cited was higher than for PW-MA reviews. Additionally, the number of times reviews achieved level one and level two impacts was higher for NMA reviews than PW-MA reviews; but no difference was found in terms of level three impact. Our findings contradict those of Lunny *et al* whose findings suggest that NMA reviews are less cited than PW-MA reviews but their work included all types of reviews rather than Cochrane Reviews alone [15].

## Strengths and weaknesses of the research

We were able to match 38 of the 60 included NMA to a PW-MA published by the same Cochrane Review Group and within a month of each other. However, there may be other confounding factors that require matching to ensure the comparison of NMA and PW-MA reviews is unbiased. We did not match additional factors because there is no research exploring which factors are significant, and, if we matched for other factors, we are likely to be able to match even fewer PW-MA reviews to NMA reviews. Ideally, the NMA and PW-MA reviews would be identical in terms of all characteristics (e.g. aims, publication date) except the

**Table 3. Guideline development groups that cited NMA reviews.**

| Guideline author | Number of guidelines |
|---|---|
| National Institute for Health and Care Excellence | 15 |
| World Health Organization | 7 |
| German Society of Neurology | 3 |
| German, Austrian, and Swiss Society of Gynaecology and Obstetrics | 3 |
| Lung Foundation Australia, Thoracic Society of Australia and New Zealand | 3 |
| Royal College of Obstetricians & Gynaecologists | 3 |
| Co-authors | 2 |
| Dutch College of General Practitioners working group. | 2 |
| European Society of Anaesthesiology and Intensive Care | 2 |
| German Society for Psychiatry and Psychotherapy, Psychosomatics and Neurology | 2 |
| Guideline Program Oncology of the Association of Scientific Scientists Medical Societies, German Cancer Society and the German Cancer Aid Foundation. | 2 |
| Queensland Maternity and Neonatal Clinical Guidelines Program | 2 |
| Russian Society of Psychiatrists | 2 |
| The Management of Chronic Obstructive Pulmonary Disease Work Group | 2 |
| The National Women and Infants Programme, Institute of Obstetricians and Gynaecologists of the Royal College of Physicians of Ireland | 2 |
| American Academy of Ophthalmology | 1 |
| American Heart Association Heart Failure and Transplantation Committee of the Council on Clinical Cardiology, Council on the Kidney in Cardiovascular Disease, Council on Cardiovascular Surgery and Anaesthesia, Council on Cardiovascular and Stroke Nursing, Council on Quality of Care and Outcomes Research, Council on Lifelong Congenital Heart Disease and Heart Health in the Young | 1 |
| American Psychiatric Association | 1 |
| Association of Ontario Midwives | 1 |
| Australian Research Centre for Population Oral Health | 1 |
| Canadian Paediatric Society Adolescent Health Committee | 1 |
| Centre for Research Excellence in Women's Health in Reproductive Life, American Society of Reproductive Medicine, Endocrine Society, European Society of Endocrinology, European Society of Human Reproduction and Embryology | 1 |
| Chronic Obstructive Pulmonary Disease Guideline Development Group | 1 |
| Clinical Practice Guideline for Hormonal and Intrauterine Contraception Working Group | 1 |
| Committee of the Taiwan Academy of Paediatric Allergy, Asthma and Immunology | 1 |
| Danish Health Authority | 1 |
| European Academy of Paediatric Dentistry | 1 |
| European Organization for Caries Research, European Federation of Conservative Dentistry, German Association of Conservative Dentistry | 1 |
| European Psychiatric Association | 1 |
| European Society for Vascular Surgery | 1 |
| European Society of Cardiology and European Respiratory Society. | 1 |
| Federation of Medical Specialists | 1 |
| Finnish Medical Society Duodecim, Finnish Angiology Association, Finnish Cardiological Society | 1 |
| Finnish Medical Society Duodecim, Finnish Association of Respiratory Physicians | 1 |
| Finnish Medical Society Duodecim, Finnish Psychiatric Association, Finnish Youth Psychiatric Association | 1 |
| Finnish Medical Society Duodecim, the Finnish Society of Anaesthesiology, the Intensive Care Medicine Subdivision and the Finnish Nephrological Society | 1 |

(*Continued*)

**Table 3.** (Continued)

| Guideline author | Number of guidelines |
|---|---|
| German Medical Association, Association of German Medical Associations, National Association of Statutory Health Insurance Physicians, Association of Scientific Medical Societies. | 1 |
| German Society for Gastroenterology, Digestive and Metabolic Diseases and the German Society of General and Visceral Surgery. | 1 |
| German Society for Psychosomatic Medicine and Medical Psychotherapy | 1 |
| German Society for Ultrasound in Medicine, German Society for Gynaecology and Obstetrics. | 1 |
| German Society of Gynaecology and Obstetrics, German Society for Midwifery Science | 1 |
| German Society of Neurology, German Stroke Society | 1 |
| Global Initiative for Chronic Obstructive Lung Disease | 1 |
| Hepatobiliary Study Group of the Chinese Society of Gastroenterology of the Chinese Medical Association, Hepatology Committee of the Chinese Research Hospital Association | 1 |
| PAPPS Women's Group | 1 |
| Practice Committee of the American Society for Reproductive Medicine | 1 |
| Registered Nurses' Association of Ontario | 1 |
| Royal Australian College of General Practitioners | 1 |
| Russian Society of Obstetricians and Gynaecologists, | 1 |
| Russian Society of Obstetricians and Gynaecologists, Association of Anaesthesiologists and Resuscitators, Association of Obstetric Anaesthesiologists-Resuscitators, National Association of Patient Blood Management Specialists | 1 |
| Society of Obstetricians and Gynaecologists of Canada | 1 |
| South African Thoracic Society | 1 |

application of the NMA or PW-MA methodology. However, this would not be feasible because such reviews are not published in CDSR.

We would expect that reviews are more likely to be cited if they address important unanswered questions that would be considered to be high priority to researchers and guideline developers. However, we did not explore whether high priority reviews were more likely to be published than low priority reviews because, to assess the priority level of each review would require collaboration of clinicians across a range of many areas and would be highly subjective and potentially controversial.

Previous versions of included reviews were excluded. Some guidelines cited previous versions of the same review, even when a new version had been published, and such guidelines were not captured in this research.

Citation in guidelines may be influenced by external triggers, the developers' resources, and updating schedules. The included reviews may still be published in future guidelines; therefore, it is possible that we have under-estimated the citation of reviews.

We included only Cochrane Reviews of interventions because guideline citation has been routinely collected by Cochrane UK and to provide a homogeneous cohort to potentially identify factors that may affect the likelihood of citation [40]. Further work is needed to evaluate the impact of other types of reviews.

## Conclusions and implications for practice

Cochrane NMA reviews appear to have more impact than PW-MA reviews, but many are not cited in guidelines. Further work is needed to explore the barriers to use of NMAs and to promote their appropriate use and interpretation. NMA involves more complex statistical analysis

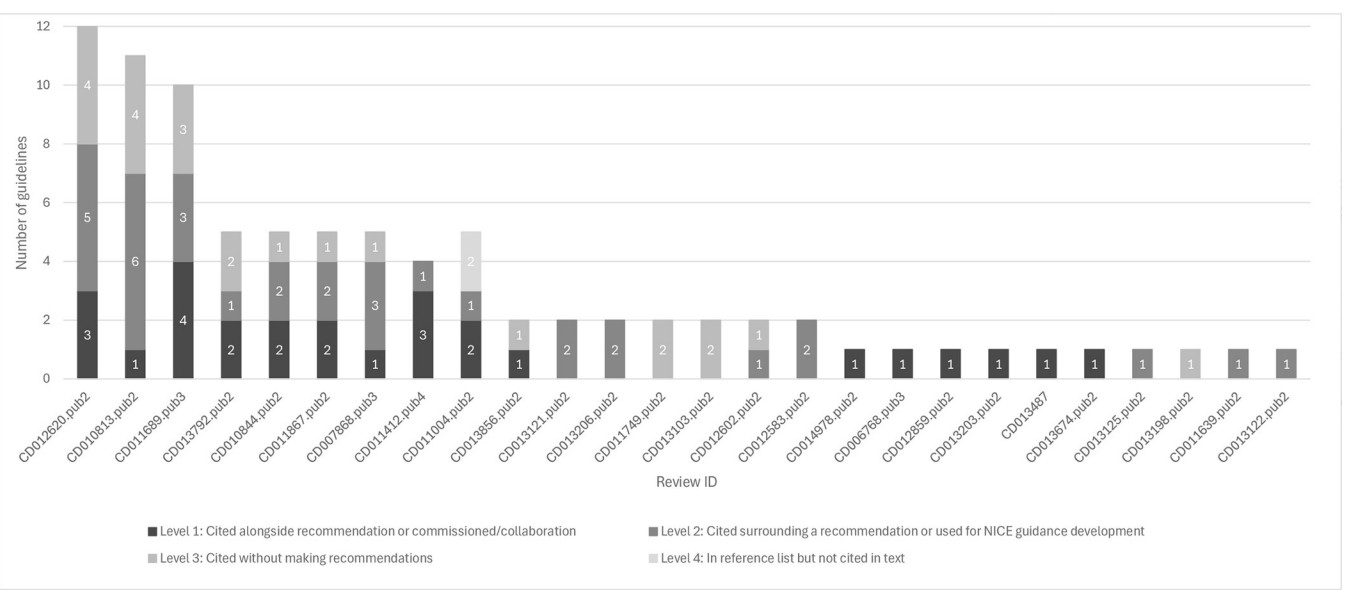

**Fig 7. Why each NMA review was used in guidelines.**

and reporting of results than PW-MA therefore lack of understanding and familiarity of the methods or results may be more problematic. Improved collaboration and communication between review authors and guideline developers will help improve research efficiency by ensuring the research question is defined appropriately, informing guideline developers of a review's existence and findings, identifying whether there are areas for which a review is needed, facilitating knowledge exchange regarding specialist topics such as NMA, and avoiding un-necessary repetition of research.

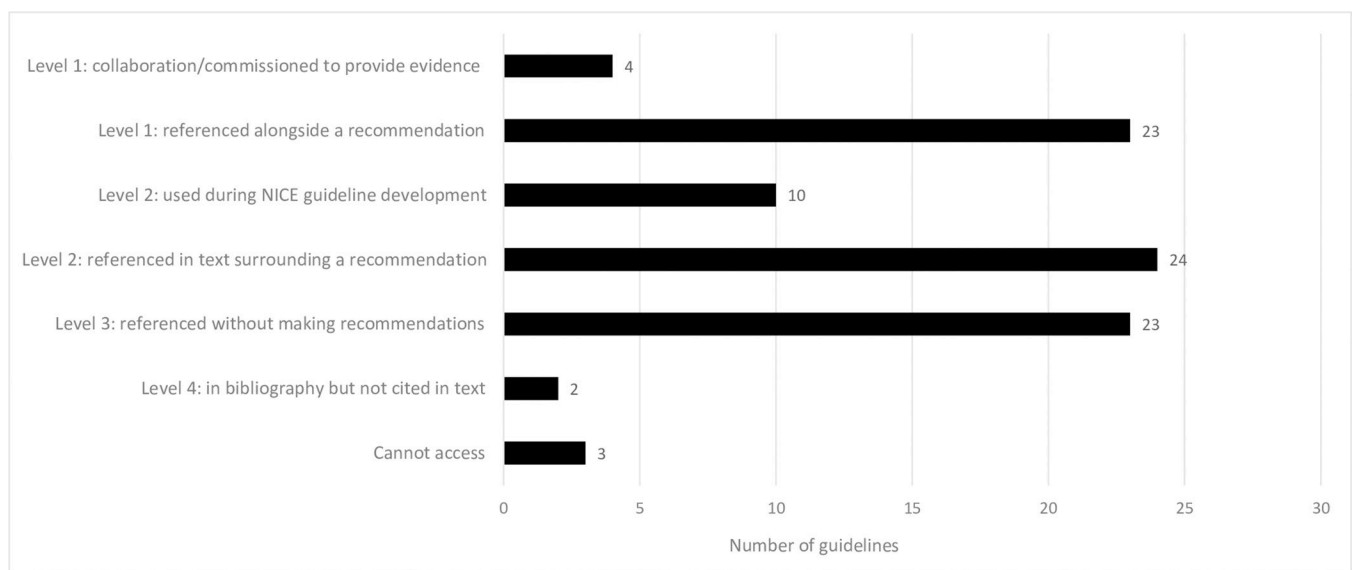

**Fig 8. Why NMA reviews were used in guidelines.**

## Supporting information

**S1 Checklist. PRISMA 2020 checklist.**
(DOCX)

**S2 Checklist. PRISMA 2020 for abstracts checklist.**
(DOCX)

**S1 File. Search strategy for Cochrane Database of Systematic Reviews.**
(PDF)

**S2 File. Eligibility assessment results.**
(PDF)

**S3 File. Matching pair-wise meta-analysis reviews to included network meta-analysis reviews.**
(PDF)

**S4 File. R code for figures and analyses.**
(PDF)

**S1 Table. Data extracted for network meta-analysis reviews that were not cited in guidelines.**
(PDF)

**S2 Table. Data extracted for network meta-analysis reviews that were cited in guidelines.**
(PDF)

**S3 Table. Data extracted for guidelines that cited network meta-analysis reviews.**
(PDF)

**S4 Table. Data extracted for pairwise meta-analysis reviews that were not cited in guidelines.**
(PDF)

**S5 Table. Data extracted for pairwise meta-analysis reviews that were cited in guidelines.**
(PDF)

**S6 Table. Data extracted for guidelines that cited pairwise meta-analysis reviews.**
(PDF)

**S7 Table. NMA and PW-MA review characteristics.**
(PDF)

**S8 Table. Characteristics of the NMAs and PW-MAs for the first outcome.**
(PDF)

**S9 Table. Guideline groups that cited NMA and PW-MA reviews.**
(PDF)

**S1 Fig. Number of NMA reviews and PW-MA reviews cited in guidelines and those not cited, for each publication year.**
(TIF)

**S2 Fig. Time to first citation in guidelines for NMA reviews and PW-MA reviews that were cited.**
(TIF)

**S3 Fig. Kaplan Meier curve of time to first citation in guidelines for NMA reviews and PW-MA reviews.**
(TIF)

**S4 Fig. Number of times NMA reviews and PW-MAs were cited in guidelines (excluding reviews that were not cited).**
(TIF)

**S5 Fig. Location of guidelines that cited NMA reviews and PW-MA reviews.**
(TIF)

**S6 Fig. Why NMA reviews and PW-MA reviews were used in cited guidelines.**
(TIF)

## Acknowledgments

We would like to thank Anne Eisinga, Therese Docherty and Emma Carter, at Cochrane UK, for locating guidelines that cite Cochrane reviews, and for providing information regarding guideline search methods that are reported in this article.

## Author Contributions

**Conceptualization:** Sarah Donegan, Zarko Alfirevic, Catrin Tudur-Smith.

**Data curation:** Sarah Donegan, James Connor.

**Formal analysis:** Sarah Donegan.

**Methodology:** Sarah Donegan, James Connor, Zarko Alfirevic.

**Supervision:** Sarah Donegan, Zarko Alfirevic, Catrin Tudur-Smith.

**Writing – original draft:** Sarah Donegan.

**Writing – review & editing:** Sarah Donegan, James Connor, Zarko Alfirevic, Catrin Tudur-Smith.

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
