## [Decision Letter · Decision Letter 0]

24 Sep 2024

PONE-D-24-27301The impact of Cochrane Reviews that apply network meta-analysis in clinical guidelines: a systematic review.PLOS ONE

Dear Dr. Donegan,

Thank you for submitting your manuscript to PLOS ONE. After careful consideration, we feel that it has merit but does not fully meet PLOS ONE’s publication criteria as it currently stands. Therefore, we invite you to submit a revised version of the manuscript that addresses the points raised during the review process.

We look forward to receiving your revised manuscript.

Kind regards,

Nishant Premnath Jaiswal, MBBS, PhD

Academic Editor

PLOS ONE

2. Thank you for stating the following in the Competing Interests section: [I have read the journal's policy and the authors of this manuscript have the following competing interests: CTS and SD were statistical editors for Cochrane Review Groups, ZA was co-ordinating editor for the Cochrane Pregnancy and Childbirth Group, and CTS is a co-convenor for the Cochrane Statistical Methods Group.]. Please confirm that this does not alter your adherence to all PLOS ONE policies on sharing data and materials, by including the following statement: "This does not alter our adherence to PLOS ONE policies on sharing data and materials.” (as detailed online in our guide for authors http://journals.plos.org/plosone/s/competing-interests). If there are restrictions on sharing of data and/or materials, please state these. Please note that we cannot proceed with consideration of your article until this information has been declared. Please include your updated Competing Interests statement in your cover letter; we will change the online submission form on your behalf.

3. As required by our policy on Data Availability, please ensure your manuscript or supplementary information includes the following:

Additional Editor Comments (if provided):

Reviewers' comments:

Reviewer's Responses to Questions

**Comments to the Author**

1. Is the manuscript technically sound, and do the data support the conclusions?

Reviewer #1: Yes

Reviewer #2: No

2. Has the statistical analysis been performed appropriately and rigorously? 

Reviewer #1: Yes

Reviewer #2: No

3. Have the authors made all data underlying the findings in their manuscript fully available?

Reviewer #1: Yes

Reviewer #2: Yes

4. Is the manuscript presented in an intelligible fashion and written in standard English?

Reviewer #1: Yes

Reviewer #2: Yes

5. Review Comments to the Author

Reviewer #1: 1. The eligibility criteria for including reviews in this study need to be spelt out well.

2. It appears that only 55% of the citations identified were screened independently. This needs explanation.

3. The rationale for comparing reviews with NMA against those with PW-MA is unclear, unless they are on the same or similar topics.

4. How were reviews with NMA and those with PW-MA ‘matched’? It appears that matching was done on the basis of publication date. It this is true, it could create an imbalance on account of the fact that the review topic(s)/methods/interventions/etc. may be so dissimilar as to make them incomparable.

5. Please remove the side-headings in the Results section.

Reviewer #2: Thank you for the opportunity to go through the wonderful article, The authors here tried to investigate the extent to which Cochrane Reviews that utilize network meta-analysis (NMA) are cited in guidelines. The study aims to explore various factors related to the citation of these reviews, including review characteristics, citation timing, location, authorship, and the specific contribution of the reviews to the guidelines. Additionally, the study compares the citation rates of NMA reviews to those of pairwise meta-analysis (PW-MA) reviews to assess the potential impact of NMA on guideline development.

The write up is good and easy to understand. I have concerns regarding the methodology –

1.A clear and exact search strategy needs to be given instead just the key terms.

2.The identified NMA reviews need to be grouped based on their field/ specific areas addressed. Every other study does not get cited in a guideline if it is not an immediate policy priority. So, an NMA review getting citation or not depends largely on if it is a priority topic at that point of time or not. This factor is a huge predictor in this study and authors have not accounted for this. Grouping NMA reviews would help in that, later a stratified analysis should be presented.

3.The procedure to match PW-MA reviews to NMAs is also a little sketchy. Was it done completely based on temporal proximity if they come from the same review group? Though they come from same review groups, the topics can be completely different, and thereby how can these be compared? If the authors matched the PW-MA reviews based on objectives, that would have been comparable.

Please clarify these questions before resubmission of the manuscript.

6. PLOS authors have the option to publish the peer review history of their article (what does this mean?). If published, this will include your full peer review and any attached files.

Reviewer #1: **Yes: **Joseph L Mathew

Reviewer #2: No

---

## [Author Response · Author response to Decision Letter 0]

4 Nov 2024

Please see attached file 'response to reviewers.docx'

---

## [Decision Letter · Decision Letter 1]

28 Nov 2024

The impact of Cochrane Reviews that apply network meta-analysis in clinical guidelines: a systematic review.

PONE-D-24-27301R1

Dear Dr. Donegan,

We’re pleased to inform you that your manuscript has been judged scientifically suitable for publication and will be formally accepted for publication once it meets all outstanding technical requirements.

Kind regards,

Nishant Premnath Jaiswal, MBBS, PhD

Academic Editor

PLOS ONE

Reviewers' comments:

Reviewer's Responses to Questions

**Comments to the Author**

1. If the authors have adequately addressed your comments raised in a previous round of review and you feel that this manuscript is now acceptable for publication, you may indicate that here to bypass the “Comments to the Author” section, enter your conflict of interest statement in the “Confidential to Editor” section, and submit your "Accept" recommendation.

Reviewer #1: (No Response)

Reviewer #2: All comments have been addressed

2. Is the manuscript technically sound, and do the data support the conclusions?

Reviewer #1: Yes

Reviewer #2: Partly

3. Has the statistical analysis been performed appropriately and rigorously? 

Reviewer #1: I Don't Know

Reviewer #2: Yes

4. Have the authors made all data underlying the findings in their manuscript fully available?

Reviewer #1: Yes

Reviewer #2: Yes

5. Is the manuscript presented in an intelligible fashion and written in standard English?

Reviewer #1: No

Reviewer #2: Yes

6. Review Comments to the Author

Reviewer #1: (No Response)

Reviewer #2: (No Response)

7. PLOS authors have the option to publish the peer review history of their article (what does this mean?). If published, this will include your full peer review and any attached files.

Reviewer #1: No

Reviewer #2: **Yes: **Anju Pradhan Sinha

---

## [Editor Report · Acceptance letter]

9 Dec 2024

PONE-D-24-27301R1 

PLOS ONE

Dear Dr. Donegan, 

I'm pleased to inform you that your manuscript has been deemed suitable for publication in PLOS ONE. Congratulations! Your manuscript is now being handed over to our production team.

Kind regards, 

on behalf of

Dr. Nishant Premnath Jaiswal 

Academic Editor

PLOS ONE